# XTT assay for detection of bacterial metabolic activity in water-based polyester polyurethane

**Nallely Magaña-Montiel**☯, **Luis Felipe Muriel-Millán**☯, **Liliana Pardo-López**📧*

Departamento de Microbiología Molecular, Instituto de Biotecnología, Cuernavaca, Morelos, México

☯ These authors contributed equally to this work.
* liliana.pardo@ibt.unam.mx

**Data Availability Statement:** All relevant data are within the paper and its Supporting Information files. dx.doi.org/10.17504/protocols.io.4rl27zbjg1y/v1.

## Abstract

Cellular metabolic activity can be detected by tetrazolium-based colorimetric assays, which rely on dehydrogenase enzymes from living cells to reduce tetrazolium compounds into colored formazan products. Although these methods have been used in different fields of microbiology, their application to the detection of bacteria with plastic-degrading activity has not been well documented. Here, we report a microplate-adapted method for the detection of bacteria metabolically active on the commercial polyester polyurethane (PU) Impranil®DLN using the tetrazolium salt 2,3-bis [2-methyloxy-4-nitro-5-sulfophenyl]-2H-tetrazolium-5-carboxanilide (XTT). Bacterial cells that are active on PU reduce XTT to a water-soluble orange dye, which can be quantitatively measured using a microplate reader. We used the *Pseudomonas putida* KT2440 strain as a study model. Its metabolic activity on Impranil detected by our novel method was further verified by Fourier-transform infrared spectroscopy (FTIR) analyses. Measurements of the absorbance of reduced XTT at 470 nm in microplate wells were not affected by the colloidal properties of Impranil or cell density. In summary, we provide here an easy and high-throughput method for screening bacteria active on PU that can be adapted to other plastic substrates.

## Introduction

Plastic pollution is one of the most serious ecological and health problems worldwide. It is estimated that there are 82 to 358 trillion plastic particles on the surface of the oceans [1], which negatively affect marine organisms and potentially human health [2]. One of the most widely produced plastic polymers is polyurethane (PU) because it is widely used in everyday products of different sectors such as healthcare, aeronautic, automotive, construction, and furniture [3], making it one of the main plastics found as contaminants in marine and terrestrial environments [4].

Among the strategies to counteract polyurethane contamination, bacterial degradation is of main interest for waste management and upcycling [5, 6]. Commonly, screening methods to detect PU-degrading bacteria are based on agar plates supplemented with PU substrates,

**Funding:** The research was funded by the PAPIIT-DGAPA UNAM program (IG200223). Funder had no role in study design, data collection and analysis, decision to publish, or preparation of the manuscript. NM-M is a doctoral student of Programa de Maestría y Doctorado en Ciencias Bioquímicas UNAM and received a Ph.D. fellowship from CONAHCYT (CVU 417065). LFM-M received a postdoctoral fellowship from CONAHCYT (CVU 296736).

**Competing interests:** The authors have declared that no competing interests exist.

where plastic degradation activity is visualized by the formation of clear zones around bacterial colonies [7–9]. However, these techniques are limited to water-based PU that is homogeneously dispersed throughout the solid media [8]. Furthermore, for marine bacterial screening, this type of method may yield false positives by the action of agarase enzymes, which also produce clear zones around the colony [10–12].

Additionally, growth in liquid cultures of bacterial strains that use plastic as the sole carbon source is often determined by turbidimetry or plate colony counting methods. Although the turbidimetric method is easy to use, it cannot distinguish between living and dead cells [13, 14]. In contrast, bacterial colony-forming unit (CFU) counts estimate the number of viable cells. However, it can be tedious and time-consuming when evaluating a large number of isolates and conditions [13]. The study of the capability of plastic biodegradation can be accelerated by the development and standardization of new, simple and easy-to-use methods for detecting environmental bacteria active on different plastic substrates and conditions, like in marine media (i.e.).

Detection of metabolically active cells can be performed by simple colorimetric methods such as the tetrazolium salt assays that use MTT (3-[4,5-dimethylthiazol-2-yl]-2,5-diphenyl tetrazolium bromide) or XTT (2,3-bis [2-methyloxy-4-nitro-5-sulfophenyl]-2H-tetrazolium-5-carboxanilide) [15, 16]. The principle of both assays is that only active cells will reduce MTT and XTT salts to formazan compounds. The amount of formazan products is directly proportional to the number of active cells and can be quantified spectrophotometrically [17]. The main difference between these assays is that MTT is converted into insoluble violet crystals (which can be dissolved with detergents or solvents), whereas XTT, which is one of the new-generation tetrazolium salts, is reduced to a soluble orange dye [18, 19].

Although tetrazolium assays were originally developed for detecting metabolic activity in eukaryotic cells [15, 16], these methods have also been used for evaluating bacterial viability under different conditions such as testing anti-mycobacterial agents [20], cytotoxic effects [21], and microbial growth [20]. Nevertheless, few studies have implemented tetrazolium-based assays for detecting bacterial activity on environmental pollutants such as phenol and hydrocarbons [22–24], and in the biodegradable polyester polycaprolactone (PCL) [25].

Here, we describe a new protocol of an XTT-based assay to detect the metabolic activity of bacteria in the presence of PU. As a proof of concept, we used the *Pseudomonas putida* KT2440 reference strain, which produced higher levels of formazan in the presence of Impranil than when it grew only with citrate as a sole carbon source. The activity on Impranil of *P. putida* KT2440 was additionally verified by Fourier transform infrared spectroscopy (FTIR) analyses. Our method was standardized in 96-well microplates, allowing its use in high-throughput screenings of many bacterial isolates and rapid detection of potential PU-degrading activity.

## Materials and methods

### Strains and culture conditions

*Pseudomonas putida* KT2440 [26–28] and *E. coli* BL21 were routinely propagated in Lysogeny Broth (LB). For XTT assays, Basal Mineral Medium (BM) was used with the following composition (in $g \cdot L^{-1}$): 0.8 $K_2HPO_4$, 0.2 $KH_2PO_4$, 0.3 $NH_4Cl$, 0.19 $Na_2SO_4$, 0.07 $CaCl_2$, 0.005 $FeSO_4 \cdot 7H_2O$, 0.16 $MgCl_2$, and 0.0002 $Na_2MoO_4$ [29] and supplemented with Instant Ocean Sea Salt (0.06 $g \cdot L^{-1}$). Water-based polyester-PU Impranil®DLN (1 $mg \cdot mL^{-1}$) from Covestro (Leverkusen, Germany) and sodium citrate (20 mM) were used as the polyurethane substrate and a simple carbon source, respectively.

## Bacterial inoculum preparation

The cryopreserved strains were reactivated in overnight liquid LB cultures incubated at 30°C and 180 rpm. Then, the pre-inocula were set as described by Oceguera-Cervantes et al. [30] with the following modifications: LB-grown cells were harvested by centrifugation at 13000 rpm, 4°C for 20 min. The cells were washed twice with 10 mL of sterile 10 mM $MgSO_4$, resuspended in 5 mL of BM, and kept on ice baths to facilitate their handling and preparation until they were used to inoculate experimental media to reach 0.1 $OD_{600nm}$. The viable cell count was verified by inoculating serial dilutions on LB agar.

## Detection of bacterial growth in polyester polyurethane

About 1 x $10^6$ washed cells were inoculated into 50-mL Erlenmeyer flasks containing 30 mL of each culture condition: BM-citrate, BM-citrate-Impranil, and BM with no carbon source. Abiotic controls (non-inoculated culture media) were also prepared, and all the cultured media were kept on ice baths. Immediately, 150 µL of each culture were pipetted and transferred to a 96-well optical-bottom plate and 50 µL of XTT (2 mg·$mL^{-1}$) was added to each well. The plate was incubated for 25 h at 30°C and 180 rpm in an automated microplate spectrophotometer reader EPOCH2 (BioTek Instruments Inc.). Optical densities (OD) at wavelengths of 470 nm and 630 nm were measured immediately after the addition of XTT and every hour of incubation. The $OD_{630}$ was used as a reference wavelength to reduce the noise of the particles and aggregates (background subtraction at 630–690 nm) dispersed in the medium [22]. The OD values at 630 nm were subtracted from subsequent readings to obtain the change in absorbance:

Absorbance corrected = [$OD_{470nm}$—$OD_{630nm}$]

The same determinations were made to the abiotic controls. The 50-mL Erlenmeyer flasks were incubated at 30°C and 180 rpm for the FTIR analyses. At least two independent experiments with three biological replicates and two technical replicates were measured. The average and the standard deviation of the absorbances were calculated for each of the evaluated conditions. Data analysis was processed with GraphPad Prism 8.0.2 (GraphPad Software) to obtain the growth curves presented in this work with averages and SD.

## FTIR analyses of Impranil

Aliquots of 2 mL from cultures in 50-mL Erlenmeyer flasks were taken at 0 and 15 days of incubation at 30°C, 180 rpm. Then, the samples were centrifuged at 4000 g x 60 s at room temperature, and the supernatants were carefully recovered and transferred to clean 2-mL tubes. To recover the remnants of Impranil, the tubes were incubated uncapped at 37°C until the supernatants evaporated [31]. To detect changes in the functional chemical groups of the dried polymer, Attenuated Total Reflection-Fourier Transform Infrared Spectroscopy (ATR-FTIR) was performed on FTIR equipment with a diamond ATR from Perkin Elmer, model Spectrum2. The spectra were acquired from 4000 to 450 $cm^{-1}$ with 0.5 $cm^{-1}$ standard resolution and 4 scans. The interpretation of the IR spectra (baseline correction and the average spectra of three biological replicates) was performed using the Spectragryph Optical Spectroscopy Software [32].

The protocol described in this peer-reviewed article is published on protocols.io, dx.doi. org/10.17504/protocols.io.4r3l27zbjg1y/v1 and is included for printing as supporting information in S1 File with this article. S2 File for the Experimental Data, and S3 File for supplementary material.

## Results

### Growth of bacterial strains in BM medium

The first step in establishing a protocol that would allow high throughput assessment of PU degradation is to select the right culture media. Because our long-term interest is to search for PU-degrading bacteria in the marine environment, we decided to use a mineral basal medium with a low concentration of marine salts (BM).

To assess bacterial growth in this medium, we selected *P. putida* KT2440 and *E. coli* BL21 strains as positive and negative controls, respectively. We chose citrate as an easily metabolized carbon source, as this organic acid promotes PU degradation in *Pseudomonas* [33]. We inoculated the bacteria at a concentration of approximately $1\text{x}10^6$ cells·mL$^{-1}$ and used 150 μL in 96-well plates.

We monitored the growth of the bacterial strains by measuring light scattering at 630 nm (OD$_{630}$). After 17 h of incubation, *P. putida* KT2440 grew in BM with citrate as the sole carbon source (Fig 1A, square symbols). In contrast, the negative control *E. coli* BL21 strain did not grow in BM supplemented with citrate (Fig 1B).

Then, we determined the growth of both strains in BM-citrate medium supplemented with 1 mg·mL$^{-1}$ Impranil. As shown in Fig 1A (diamond symbols), *P. putida* KT2440 exhibited a shorter lag phase in the presence of Impranil than when cultured with citrate alone, suggesting that the *P. putida* KT2440 strain metabolized Impranil (However, both cultures of *P. putida* KT2440 reached a similar OD$_{630}$ at the end of the kinetic.). In contrast, the addition of Impranil did not improve the growth of *E. coli* BL21 (Fig 1B).

However, we noticed that wells containing *P. putida* KT2440 cultures with Impranil showed particle formation (Fig 2C–3), which could increase multiple times the scattering of the incident light in wells (multiple scattering regime) [14] and therefore overestimate the bacterial growth. We also observed aggregate formation in *P. putida* KT2440 cultures in the 50-mL Erlenmeyer flasks containing BM-citrate-Impranil medium (S2 Fig in S3 File). No changes were observed in the turbidity of the culture media for abiotic controls (Fig 2A). Similarly, no

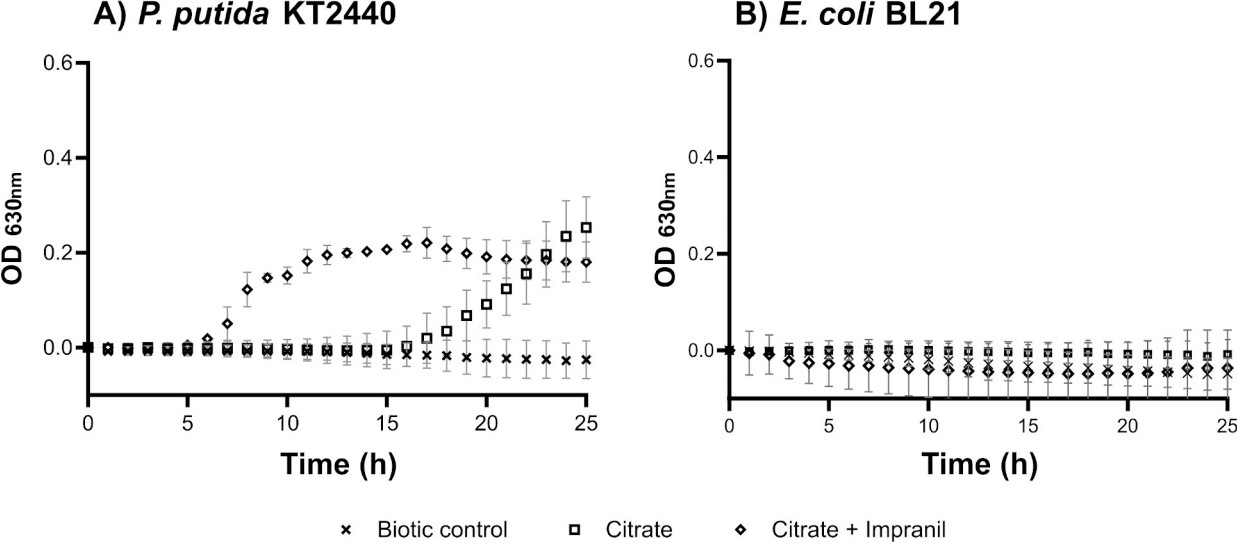

**Fig 1. Growth curves of bacteria in BM medium based on OD630.** A) P. putida KT2440 and B) E. coli BL21 were grown in BM medium with either 20 mM citrate or 20 mM citrate and 1 mg·mL$^{-1}$ Impranil for 25 hours at 30°C, 180 rpm in a microplate reader. Biotic controls correspond to bacterial cultures without any carbon source. The data are the mean of three independent experiments performed in duplicate. Error bars indicate the standard deviation (SD).

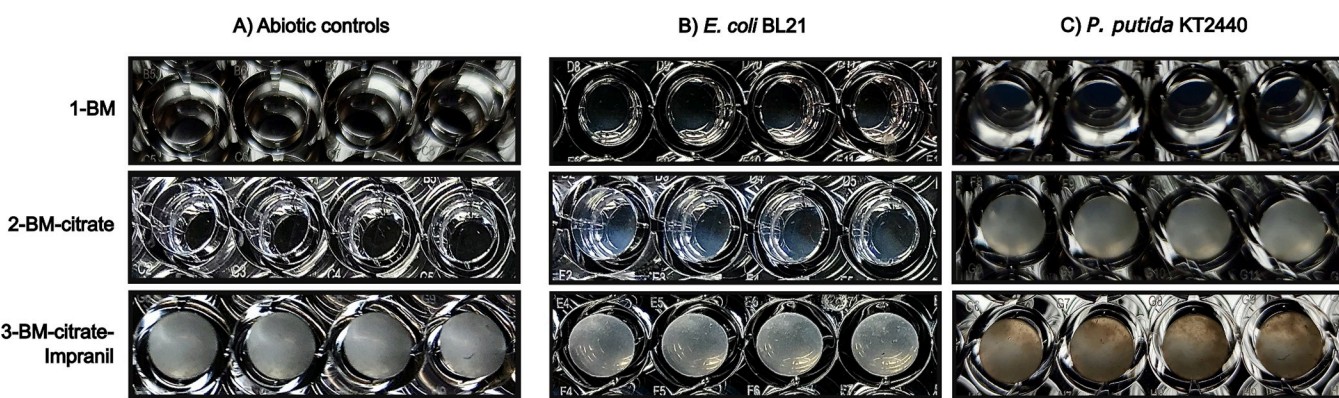

**Fig 2.** *putida* **KT2440 grown in BM-citrate-Impranil tend to form aggregates (see C-3).** *P.* Photographs of microplate wells containing cultures of P. putida KT2440, and *E. coli* in BM medium added with either citrate or citrate-Impranil after 24 h of incubation at 30°C, 180 rpm. Differences in the color of cultures in wells are due to the presence of Impranil (compare 2-BM-citrate with 3-BM-citrate-Impranil), *E. coli* BL21 inocula, or the growth of *P. putida* KT2440 using citrate or citrate-Impranil as carbon sources.

changes were observed for E. coli BL21 which is incapable of growing in citrate or Impranil as carbon source (Fig 2B).

Based on the results, the use of optical density to monitor bacterial growth with Impranil in microwell plates has low reliability, making it difficult to distinguish the contribution of the additional carbon source (in this case Impranil) in microbial development. To solve this, we propose the use of XTT assay to monitor the metabolic activity as an indicator of bacterial growth of *P. putida* KT2440 in Impranil, as shown below.

## XTT assay for detection of bacterial metabolic activity in polyester polyurethane

Once we established that *P. putida* KT2440 could grow in BM and use the PU substrate Impranil, we wanted to determine its metabolic activity by using XTT reduction to colored formazan. We added 50 μL of XTT (0.5 mg·mL$^{-1}$ final concentration) to each well of the plates. After 25 h at 30°C, the plate wells corresponding to the abiotic controls, the inoculated culture media without carbon source, and those with *E. coli* BL21 cultures remained colorless (Fig 3A, 3B and 3C-1). In contrast, wells containing *P. putida* KT2440 grown on BM-citrate and BM-

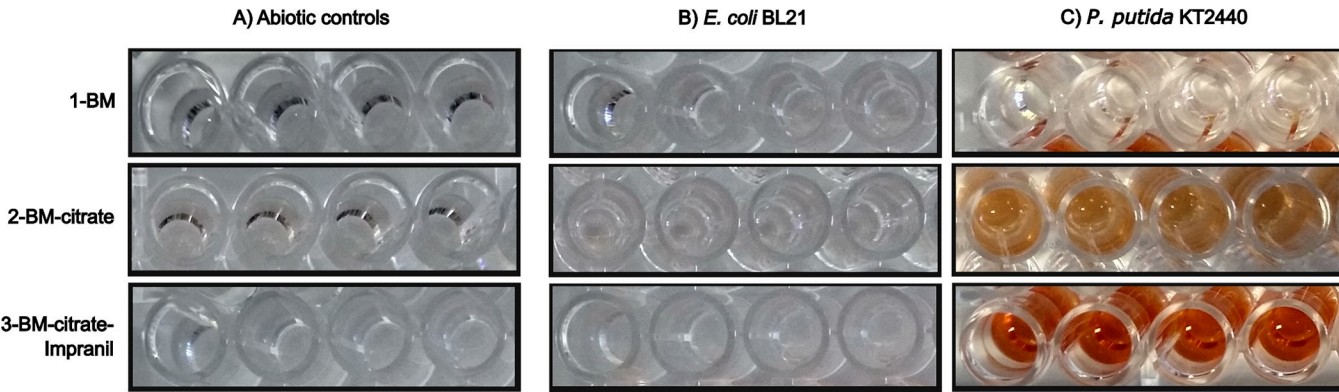

**Fig 3. Metabolically active P. putida KT2440 cells reduce the XTT salt to a soluble, orange-colored formazan.** Photographs of microplate wells containing cultures of *P. putida* KT2440 and *E. coli* grown in BM medium added with either citrate or citrate-Impranil and XTT after 25 hours of incubation. The intensity of the orange formazan is directly proportional to the number of active cells. *E. coli* BL21 negative control and abiotic controls did not show XTT reduction.

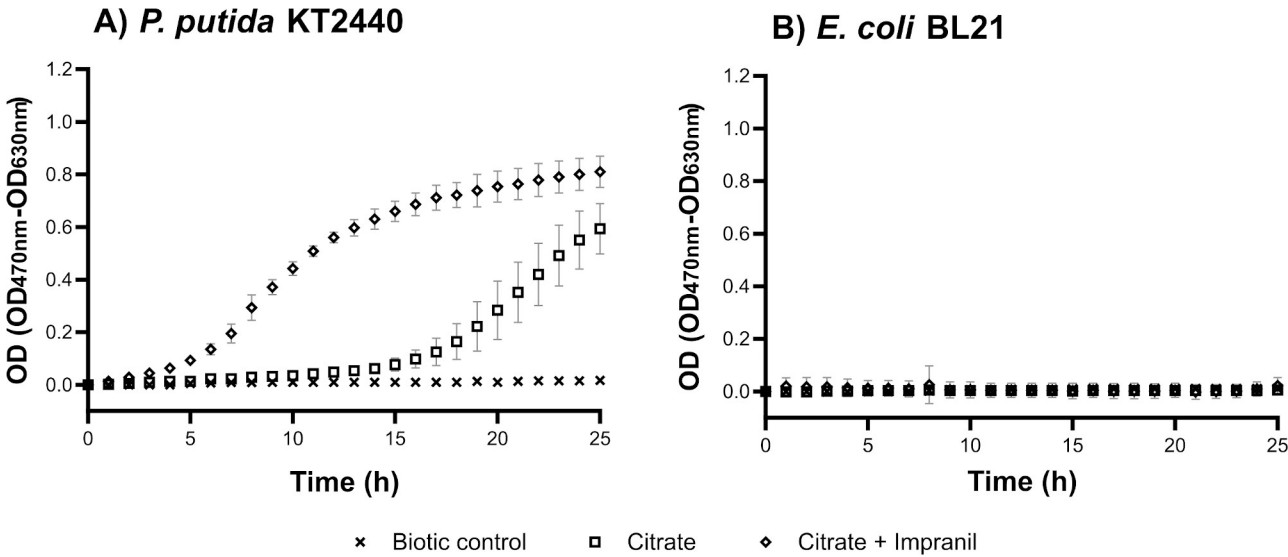

**Fig 4. Kinetics of orange-colored formazan production based on OD470.** 50 μL of XTT (2 mg·mL$^{-1}$) were added to the cultures in microplates of **(A)** *P. putida* KT2440 and **(B)** *E. coli* BL21 described in the legend of Fig 1. The data are the mean of the corrected absorbance (OD470-OD630) of three independent experiments performed in duplicate. Error bars; SD.

citrate-Impranil showed an orange color (Fig 3C–2 and 3C-3), consistent with the reduction of XTT to formazan by active cells. Furthermore, the intensity of the orange color in wells containing *P. putida* KT2440 grown in BM-citrate-Impranil suggests that the strain was more active in the presence of PU than when it grew with citrate alone.

To quantify the metabolic activity of *P. putida* KT2440 in BM-citrate-Impranil, we performed the kinetics of formazan production by measuring the OD at 470 nm (OD$_{470}$). The OD$_{630}$ values were subtracted from those at OD$_{470}$ to reduce the noise caused by aggregates in the culture media (see Materials and methods and the set of raw data and calculations in S2 File). The kinetics of formazan production in *P. putida* KT2440 cultures supplemented with either citrate or citrate and Impranil exhibited similar behavior to those of light scattering measurements (Figs 1 and 4), in which the presence of Impranil increased the metabolic activity of the strain.

However, the kinetics of formazan production of *P. putida* KT2440 grown with Impranil showed a curve with a better sigmoid shape and less variation than that obtained by light scattering measurement (Fig 4A). Furthermore, the measurement of XTT formazan production allowed us to determine that, after 25 hours of incubation, *P. putida* KT2440 had higher metabolic activity in the presence of Impranil than when grown with citrate alone. As we observed in previous OD630 measurements, the addition of Impranil did not improve the growth of *E. coli* BL21, we then did not detect the production of XTT formazan (Fig 4B). Altogether, these results demonstrate that XTT reduction is a reliable method for detecting bacteria that are metabolically active in PU.

## FTIR analysis of Impranil

To further demonstrate the activity of *P. putida* KT2440 on Impranil, we analyzed the dried polymer recovered from cultures at 0 and 15 days of incubation (30°C and 180 rpm) by FTIR spectroscopy. The spectrum of Impranil incubated with *P. putida* KT2440 for 15 days showed several changes compared to that of day 0 (Fig 5) and to that of the abiotic control (S3 Fig in S3 File). We identified the characteristic functional groups for Impranil in the FTIR spectra

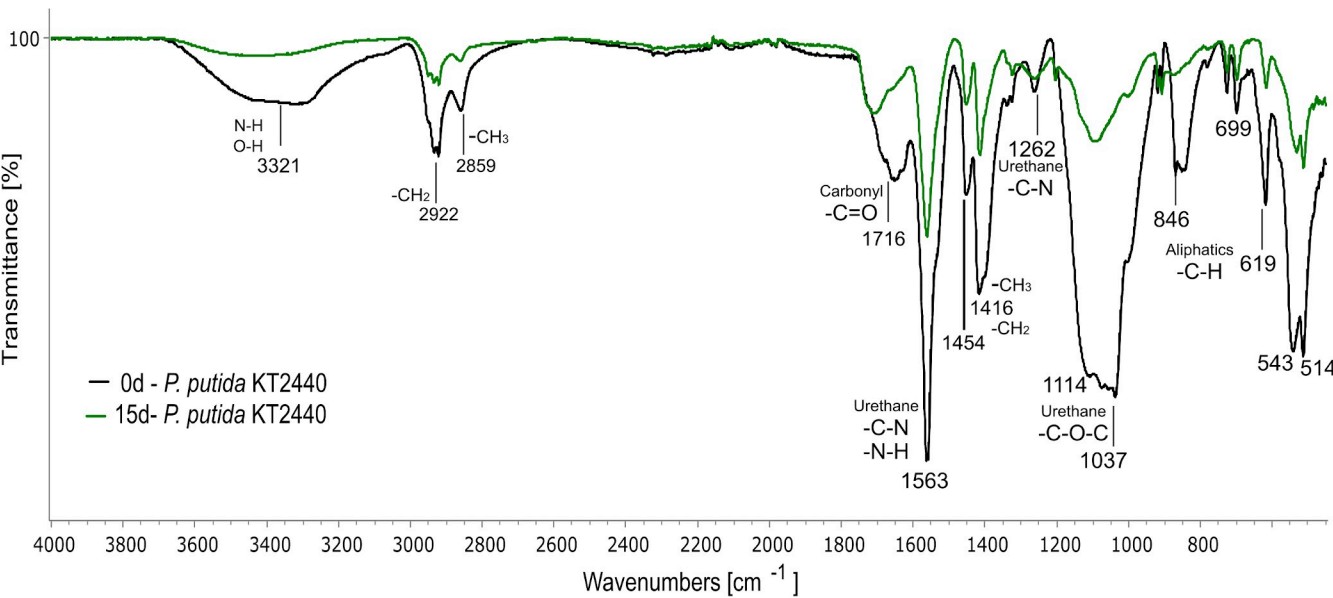

**Fig 5. FTIR spectra of Impranil treated with *P. putida* KT2440 for 0 and 15 days of incubation.** Changes in functional groups are observed. The FTIR spectra show the average of three biological replicates for each sampling day averaged with Spectragryph software.

(Fig 5): the absorption peak at 3321 cm$^{-1}$ in the *P. putida* KT2440 sample for day zero is characterized by the N-H and O-H stretching movement, and the peaks at 2922 and 2859 cm$^{-1}$ correspond to the C-H stretching of methylene and methyl groups, respectively. The C = O stretching vibration (peak 1706 cm$^{-1}$) represents the carbonyl ester functional group in Impranil, the peaks at 1563 cm$^{-1}$ correspond to the N-H bending of urea or urethane added to the C-N stretching signal, the peak at 1416 and 1454 cm$^{-1}$ are associated with the -CH$_2$ bond, the C-N stretching at 1262 cm$^{-1}$, and the C-O-C stretching of urethane at 1037 cm$^{-1}$. The signals in the last part of the spectrum (far right) usually correspond to aliphatic chains (846, 699, 619 cm$^{-1}$) [31, 34–36]. The spectrum of the *P. putida* KT2440 strain treated with Impranil for 15 days revealed a significant decrease in the intensity of the carbonyl (1706 cm$^{-1}$) and urethane (1037 cm$^{-1}$) signals, which are related to enzymatic activity on the ester group of PU [36–38]. FTIR spectra of abiotic controls and *E. coli* BL21 did not show significant changes over time (S3 and S4 Figs in S3 File).

In sum, these results in FTIR experiments confirm the active metabolism of *P. putida* KT2440 on Impranil detected by the XTT assay, and suggest that this strain can use this polymer as a carbon source under our experimental conditions.

## Discussion

The present study proposes an easy colorimetric method to detect bacteria metabolically active on PU. Our assay is based on the action of bacterial respiratory enzymes on the XTT compound to produce a soluble orange formazan that is directly proportional to the number of live cells. This is an advantage over turbidimetric assays (e.g., OD$_{600}$) because inactive or dead cells do not interfere with the formazan measurements [19].

Screening for PU degradation using agar plate-based methods is widely used and has allowed the identification of bacterial strains with high degrading activity [8, 39, 40]. However, a negative phenotype in agar plate screening does not necessarily indicate the absence of PU-degrading activity. For instance, Su et al. [41] reported that a bacterial consortium that did not

show clear zones on Impranil agar plates could cause chemical and physical changes to PU films and generated monomers of adipic acid and butanediol, which are degradation products of PU. These discrepancies could be related to the low dynamic range of the agar plate screenings or the inability to visualize weak signals [42].

Similarly, the microorganism model used here, the reference strain *P. putida* KT2440, was previously reported to be unable to produce clearing halos on Impranil agar plates with and without citrate [8, 33]. However, we determined that the metabolism of this strain was increased in the presence of PU (Fig 4), and FTIR analysis confirmed that *P. putida* KT2440 produced chemical changes on Impranil, which are compatible with PU degradation (Fig 5). These findings and previous evidence that *P. putida* KT2440 can use butanediol as a carbon source [28] support that this strain is capable of degrading PU. Nevertheless, enzymes of *P. putida* KT2440 active on PU remain to be known as this strain does not have homologs of polyurethanases from other *Pseudomonas* species [33].

Previous studies have reported the detection of bacterial metabolic activity on plastics and aromatic compounds using tetrazolium salts such as 2,3,5-Triphenyltetrazolium (TTC) and MTT [25, 24], which are reduced to insoluble formazan derivatives [43]. This limits direct quantitative assessments because previous solvent solubilization of these compounds is required. In contrast, the XTT-based assay reported here allows the quantification of the reduced formazan directly from the microplate used, saving time and materials.

High-throughput screening is often adapted to the microplate format because it allows the simultaneous testing of many samples and multimode measurements. However, the evaluation of bacterial growth based on microplate OD measurement can be affected by multiple light scattering resulting from high cell densities in the wells ($OD_{600} > 0.200$) [44]. Although time derivatives of the OD curves and calibration procedures can be implemented to address this issue [44, 45], screening for bacterial growth in the presence of solid substrates or colloidal suspensions (e.g., Impranil) in microplates may not be feasible because they interfere with $OD_{600}$ measurements [45]. In contrast, our assay for detecting the bacterial metabolic activity on PU in microplates is reliable as the colloidal properties of Impranil and aggregate apparition do not affect measurements of the reduced formazan at 470 nm. Furthermore, the XTT-based assay can detect bacterial metabolic activity on plastic substrates other than Impranil (S5 Fig in S3 File). The method proves its potential for the screening of a large number of isolates and plastic substrates; even with substrates that are insoluble in water (as PCL), in combination with easily degradable carbon sources to promote growth i.e. glucose, succinate, pyruvate, or in this case citrate [22].

In conclusion, we propose a simple, direct, and high-throughput method for the detection of metabolically active bacteria on PU. The XTT-based assay can complement agar-based screenings, especially in cases of bacterial isolates with no or weak signals of clearing halo. In addition, our method can be adapted to other plastic substrates by adjusting several parameters, such as formazan salt concentration, culture medium, and incubation conditions. Nevertheless, plastic biodegradation needs to be confirmed by implementing other methods, including electron microscopy, FTIR, identification of degradation products, and detection of incorporation of plastic-derived carbon into the bacterial metabolism by stable isotope analysis.

## Supporting information

**S1 File. XTT protocol.** PDF file that contains step by step how to perform this experiment and tips for adapting it to other culture conditions. The protocol described in this peer-reviewed article, published on protocols.io, dx.doi.org/10.17504/protocols.io.4r3l27zbjg1y/v1, included

for printing.
(PDF)

**S2 File. Experimental data.** Excel file that contains the ODs readings and calculations for the determinations of bacterial metabolic activity with XTT in this protocol.
(XLSX)

**S3 File. Supplementary material.** PDF file with additional figures.
(PDF)

## Acknowledgments

We thank Dr. Guadalupe Espín and Dr. Victor Bustamente from Instituto de Biotecnología (IBt) UNAM for providing the strain *P. putida* KT2440 and his permission to use the microplate reader, respectively. We also thank Dr. Herminia Loza Tavera from Facultad de Química, UNAM and Dr. Ayixon Sánchez Reyes (IBt) for their technical support and critical suggestions.

## Author Contributions

**Conceptualization:** Nallely Magaña-Montiel.

**Formal analysis:** Nallely Magaña-Montiel, Luis Felipe Muriel-Millán, Liliana Pardo-López.

**Funding acquisition:** Liliana Pardo-López.

**Methodology:** Nallely Magaña-Montiel.

**Project administration:** Liliana Pardo-López.

**Resources:** Liliana Pardo-López.

**Supervision:** Luis Felipe Muriel-Millán, Liliana Pardo-López.

**Writing – original draft:** Nallely Magaña-Montiel, Luis Felipe Muriel-Millán.

**Writing – review & editing:** Nallely Magaña-Montiel, Luis Felipe Muriel-Millán, Liliana Pardo-López.

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
