## [Decision Letter · Decision Letter 0]

29 Feb 2024

PONE-D-24-04579XTT Assay for Detection of Bacterial Metabolic Activity in Water-based Polyester Polyurethane in marine salts.PLOS ONE

Dear Dr. Pardo-López,

Thank you for submitting your manuscript to PLOS ONE. After careful consideration, we feel that it has merit but does not fully meet PLOS ONE’s publication criteria as it currently stands. Therefore, we invite you to submit a revised version of the manuscript that addresses the points raised during the review process.

We look forward to receiving your revised manuscript.

Kind regards,

Bashir Sajo Mienda, PhD

Academic Editor

PLOS ONE

Journal Requirements:

2.When completing the data availability statement of the submission form, you indicated that you will make your data available on acceptance. We strongly recommend all authors decide on a data sharing plan before acceptance, as the process can be lengthy and hold up publication timelines. Please note that, though access restrictions are acceptable now, your entire data will need to be made freely accessible if your manuscript is accepted for publication. This policy applies to all data except where public deposition would breach compliance with the protocol approved by your research ethics board. If you are unable to adhere to our open data policy, please kindly revise your statement to explain your reasoning and we will seek the editor's input on an exemption. Please be assured that, once you have provided your new statement, the assessment of your exemption will not hold up the peer review process.

Reviewers' comments:

Reviewer's Responses to Questions

**Comments to the Author**

1. Does the manuscript report a protocol which is of utility to the research community and adds value to the published literature?

Reviewer #1: Yes

Reviewer #2: Yes

2. Has the protocol been described in sufficient detail?

To answer this question, please click the link to protocols.io in the Materials and Methods section of the manuscript (if a link has been provided) or consult the step-by-step protocol in the Supporting Information files.

The step-by-step protocol should contain sufficient detail for another researcher to be able to reproduce all experiments and analyses.

Reviewer #1: Yes

Reviewer #2: Yes

3. Does the protocol describe a validated method?

Reviewer #1: Yes

Reviewer #2: Yes

4. If the manuscript contains new data, have the authors made this data fully available?

Reviewer #1: Yes

Reviewer #2: Yes

**5. Is the article presented in an intelligible fashion and written in standard English?**

Reviewer #1: Yes

Reviewer #2: Yes

6. Review Comments to the Author

Reviewer #1: In this study, a microplate-adapted method for the detection of bacteria metabolically active on the commercial polyester polyurethane (PU) Impranil®DLN using the tetrazolium salt 2,3-bis [2- methyloxy-4-nitro-5-sulfophenyl]-2H-tetrazolium-5-carboxanilide (XTT) was developed. Bacterial cells that are active on PU reduce XTT to a water-soluble orange dye, which can be quantitatively measured using a microplate reader. The topic is quite interesting, but the explanation is not clear enough. Here give some suggestions for further revisions.

1. “In summary, we provide here an easy and high-throughput method for screening bacteria active on PU that can be adapted to other plastic substrates.”, only one type plastic substrate of Impranil was used in this study, and is it possible to provide the effect results of bacteria on other plastic substrates?

2. There is no description of Figures 2A and 2B in the paper.

3. In Figure 2B, the wells containing E.coli.BL21 cultures without and with Impranil showed light orange, why?

4. In Figure 2C, the wells containing P. putida KT2440 cultures (2-BM-citrate) showed blue and light orange, respectively, why?

5. The OD630 was used as a reference wavelength to reduce the noise of the particles and aggregates, why choose it?

Reviewer #2: The manuscript is well planned and executed. Some of the concern raised to be addressed before accepting

1. The figure 2C seems to differ from the figure shown in supplementary file 1

2. Statistical analysis details are not mentioned in the methods

3. supplementary file-2 is not getting opened

4. The color of the colony in 2C differs slightly from the rest. Is it because of the combination of Impranil and bacteria?

5. Line number 111 page number -6 20mM to be written as 20 mM

6.Page number 6, line number -119 10mM MgSO4 to be written properly

7. Many places hyphens are used eg. 2-mL , 50-mL (Page number 7, line number -142) The manuscript to be rechecked for these mistakes including superscripts, spaces and hyphens.

8. Legends of the figures may be included at the end as they obstruct the smooth reading of the manuscript

7. PLOS authors have the option to publish the peer review history of their article (what does this mean?). If published, this will include your full peer review and any attached files.

Reviewer #1: No

Reviewer #2: No

---

## [Author Response · Author response to Decision Letter 0]

12 Apr 2024

Reviewer #1: 

In this study, a microplate-adapted method for the detection of bacteria metabolically active on the commercial polyester polyurethane (PU) Impranil®DLN using the tetrazolium salt 2,3-bis [2- methyloxy-4-nitro-5-sulfophenyl]-2H-tetrazolium-5-carboxanilide (XTT) was developed. Bacterial cells that are active on PU reduce XTT to a water-soluble orange dye, which can be quantitatively measured using a microplate reader. The topic is quite interesting, but the explanation is not clear enough. Here give some suggestions for further revisions.

1. “In summary, we provide here an easy and high-throughput method for screening bacteria active on PU that can be adapted to other plastic substrates.”, only one type plastic substrate of Impranil was used in this study, and is it possible to provide the effect results of bacteria on other plastic substrates?

Ans: We have included an additional experiment in which we tested two other polyester substrates by using the XTT assay: 

1) Polycrylic (another commercial dispersion of polyester polyurethane); and 

2) polycaprolactone (PCL), which is a biodegradable water-insolublle polyester 

We found that P. putida KT2440 exhibited higher metabolic activity in the presence of either Polycrylic or PCL than in the condition of citrate as the sole carbon source. The results obtained are shown in Fig S5 in the supplementary material, and described in lines 320-324.

2. There is no description of Figures 2A and 2B in the paper.

Ans: We appreciate this observation, and figures 2A and 2B are now described in lines 192-194 in the revised manuscript.

3. In Figure 2B, the wells containing E.coli.BL21 cultures without and with Impranil showed light orange, why?

Ans. The differences in colors in wells containing E. coli cultures are due to the presence of the plastic substrate Impranil, which is a white suspension. This information is included in the legend of Figure 2 in the revised manuscript (lines 199-201). 

4. In Figure 2C, the wells containing P. putida KT2440 cultures (2-BM-citrate) showed blue and light orange, respectively, why?

Ans: The blue color in the image is due to the transparency of the culture media (2-BM-citrate in the abiotic control, or with BL21, which cannot grow with citrate); the angle and illumination at the time the image was taken makes the bottom of the well blueish. In contrast, P. putida KT2440 can use citrate as a carbon source and the presence of cells in suspension makes the media opaquer. An explanation of the variations in the color of cultures in wells in the legend of Figure 2 has been added. 

5. The OD630 was used as a reference wavelength to reduce the noise of the particles and aggregates, why choose it?

Ans: The use of a reference wavelength can significantly reduce the noise from solid particles in microwell plate assays as has been previously reported (Johnsen et al, 2002; Chileshe et al, 2019-https://doi.org/10.1177/1040638719843955). In our protocol, we used the OD630 as the reference wavelength to reduce the noise from Impranil particles and cellular aggregates caused by the P. putida KT2440 strain during measurements of the reduced formazan at 470 nm (Fig 2C). This is mentioned in lines 128-132.

Reviewer #2: 

The manuscript is well planned and executed. Some of the concern raised to be addressed before accepting.

1. The figure 2C seems to differ from the figure shown in supplementary file 1

Ans: Supplementary File 1 describes the XTT lab protocol (step by step) available at dx.doi.org/10.17504/protocols.io.4r3l27zbjg1y/v1. The figure 2C corresponds to the photograph of the microwell plate at 630 nm. 

 In lines 192-194, we mention that Fig S2 (S3 supplementary file) corresponds to photographs of cultures in 50-mL Erlenmeyer flasks to show that aggregate formation also occurs in these cultures. We have added photographs of flasks taken from their front in figure S2 and clarified in the figure legend that the supplementary material was corrected for more clarity to the reader. 

2. Statistical analysis details are not mentioned in the methods

Ans: A description of the statistical analysis is included in lines 136-138.

3. supplementary file-2 is not getting opened

Ans: We apologize for the inconvenience. We have verified that supplementary file-2 can be downloaded from the PLOS ONE submission system, and it can be opened. We have communicated this issue to the editor. 

4. The color of the colony in 2C differs slightly from the rest. Is it because of the combination of Impranil and bacteria?

Ans: Yes, it is. When using BM-citrate, the media becomes more opaque due to saturation. However, when using BM-citrate-Impranil, the color becomes more brownish due to the growth of floculli and cells.

5. Line number 111 page number -6 20mM to be written as 20 mM

Ans: We did the correction (Line 109).

6.Page number 6, line number -119 10mM MgSO4 to be written properly

Ans: We did the correction (Line 115) .

7. Many places hyphens are used eg. 2-mL , 50-mL (Page number 7, line number -142) The manuscript to be rechecked for these mistakes including superscripts, spaces and hyphens.

Ans: Thanks, it has already been corrected.

8. Legends of the figures may be included at the end as they obstruct the smooth reading of the manuscript

Ans: According to the PLOS ONE submission guidelines, figure legends should be placed immediately after the paragraph in which they are first referenced.

---

## [Decision Letter · Decision Letter 1]

22 Apr 2024

XTT Assay for Detection of Bacterial Metabolic Activity in Water-based Polyester Polyurethane in marine salts.

PONE-D-24-04579R1

Dear Dr. PARDO-LOPEZ,

We’re pleased to inform you that your manuscript has been judged scientifically suitable for publication and will be formally accepted for publication once it meets all outstanding technical requirements.

Kind regards,

Bashir Sajo Mienda, PhD

Academic Editor

PLOS ONE

Additional Editor Comments (optional):

Reviewers' comments:

Reviewer's Responses to Questions

**Comments to the Author**

1. Does the manuscript report a protocol which is of utility to the research community and adds value to the published literature?

Reviewer #1: Yes

Reviewer #2: Yes

2. Has the protocol been described in sufficient detail?

To answer this question, please click the link to protocols.io in the Materials and Methods section of the manuscript (if a link has been provided) or consult the step-by-step protocol in the Supporting Information files.

The step-by-step protocol should contain sufficient detail for another researcher to be able to reproduce all experiments and analyses.

Reviewer #1: Yes

Reviewer #2: Yes

3. Does the protocol describe a validated method?

Reviewer #1: Yes

Reviewer #2: Yes

4. If the manuscript contains new data, have the authors made this data fully available?

Reviewer #1: Yes

Reviewer #2: Yes

**5. Is the article presented in an intelligible fashion and written in standard English?**

Reviewer #1: Yes

Reviewer #2: Yes

6. Review Comments to the Author

Reviewer #1: The reviewers' comments have been well addressed, and the manuscript has been carefully revised. This manuscript can be accepted for publication in the current form.

Reviewer #2: Authors have included all the suggestions. Manuscript seems fine to me. may be accepted for publication

7. PLOS authors have the option to publish the peer review history of their article (what does this mean?). If published, this will include your full peer review and any attached files.

Reviewer #1: No

Reviewer #2: No

---

## [Editor Report · Acceptance letter]

9 May 2024

PONE-D-24-04579R1 

PLOS ONE

Dear Dr. Pardo-López, 

I'm pleased to inform you that your manuscript has been deemed suitable for publication in PLOS ONE. Congratulations! Your manuscript is now being handed over to our production team.

Kind regards, 

on behalf of

Dr. Bashir Sajo Mienda 

Academic Editor

PLOS ONE